# Tuning lithium-peroxide formation and decomposition routes with single-atom catalysts for lithium–oxygen batteries

Li-Na Song[1,7], Wei Zhang[2,3,7], Ying Wang[4,7], Xin Ge[2], Lian-Chun Zou[1], Huan-Feng Wang[1], Xiao-Xue Wang[1], Qing-Chao Liu[5], Fei Li[1] & Ji-Jing Xu [1,6✉]

Lithium-oxygen batteries with ultrahigh energy density have received considerable attention as of the future energy storage technologies. The development of effective electrocatalysts and a corresponding working mechanism during cycling are critically important for lithium-oxygen batteries. Here, a single cobalt atom electrocatalyst is synthesized for lithium-oxygen batteries by a polymer encapsulation strategy. The isolated moieties of single atom catalysts can effectively regulate the distribution of active sites to form micrometre-sized flower-like lithium peroxide and promote the decomposition of lithium peroxide by a one-electron pathway. The battery with single cobalt atoms can operate with high round-trip efficiency (86.2%) and long-term stability (218 days), which is superior to a commercial 5 wt% platinum/carbon catalyst. We reveal that the synergy between a single atom and the support endows the catalyst with excellent stability and durability. The promising results provide insights into the design of highly efficient catalysts for lithium-oxygen batteries and greatly expand the scope of future investigation.

[1] State Key Laboratory of Inorganic Synthesis and Preparative Chemistry, College of Chemistry, Jilin University, 130012 Changchun, PR China. [2] Electron Microscopy Center, School of Materlals Science and Engneering, Jilin University, 130012 Changchun, PR China. [3] IKERBASQUE, Basque Foundation for Science, 48013 Bilbao, Spain. [4] State Key Laboratory of Rare Earth Resource Utilization, Changchun Institute of Applied Chemistry Chinese Academy of Sciences, 130022 Changchun, PR China. [5] College of Chemistry and Molecular Engineering, Zhengzhou University, 100 Science Road, 450001 Zhengzhou, PR China. [6] International Center of Future Science, Jilin University, 130012 Changchun, PR China. [7] These authors contributed equally: Li-Na Song, Wei Zhang, Ying Wang. ✉email: jijingxu@jlu.edu.cn

Rechargeable lithium–oxygen (Li–O₂) batteries based on the reversible formation and decomposition of $Li_2O_2$ provide a theoretical specific energy density (3500 Wh kg$^{-1}$) that is 5–10 times higher than that of conventional Li-ion batteries, thus showing appreciable potential for vehicle applications[1–6]. A typical nonaqueous Li–O₂ battery consists of a Li-metal anode, Li⁺-conducting electrolyte, and porous oxygen cathode. During discharge, O₂ is preferentially reduced and then combines with Li⁺ to form $Li_2O_2$ (the oxygen reduction reaction, ORR). The $Li_2O_2$ is subsequently converted back into Li and O₂ during charging (the oxygen evolution reaction, OER). The electrocatalyst can effectively enhance the kinetics of both the ORR and OER, further improving the round-trip efficiency, which is crucial to energy storage equipment. In response, extensive research efforts have been made to develop highly active catalysts to boost the ORR and OER kinetics, such as the use of carbonaceous materials[7–9], transition-metal oxides[10–12], noble metals[13–15], and perovskites[16–18]. In fact, most of the aforementioned catalysts can be widely used to reduce the large overpotential; however, they have severe, unavoidable shortcomings, such as their high cost and scarcity along with their poor durability and relatively low capacity. Therefore, developing highly active and stable non-precious metal catalysts for Li–O₂ batteries is significantly important. Moreover, the deposition/decomposition route of the $Li_2O_2$ discharge products is also an important factor in determining the kinetics of the ORR and OER reactions. Many different models for the $Li_2O_2$ growth mechanism have been proposed, such as a surface-adsorption pathway and a solvation-mediated pathway, which are controlled by the DN value of the electrolyte[19], adsorption energy of the cathode to intermediate species[5], cluster size[20] and facet engineering of the catalyst[21]. All mentioned studies suggest that the solubility of $LiO_2$ intermediates in the electrolyte determines the deposition behaviour of $Li_2O_2$. However, little is known about the effective decomposition mechanism after recharging by an enhanced solvation-mediated pathway, which becomes a large hindrance of the round-trip efficiency and cycle life of the battery. As a result, adopting an effective electrocatalyst to discuss in detail the mechanism of O₂ reduction to $Li_2O_2$ while discharging and the reverse process while charging will be an another way to design high round-trip efficiency in Li–O₂ batteries.

Heterogeneous single-atom catalysts (SACs) referring to atomically dispersed active metal centres on a support have attracted considerable interest in recent years. The attention is because of the special electronic structure and maximized atomic utilization of such materials, which differ greatly from those of conventional nano- or subnano-counterparts[22–24]. Moreover, the single-atom nature of the active centres in SACs and the resulting low-coordination environment and the enhanced metal-support interactions all provide robust catalytic performance in a number of heterogeneous reactions, such as the carbon dioxide reduction[24,25], hydrogen evolution reaction[26,27] and photo-catalytic reactions[28,29]. In principle, it is vital to explore the electrochemistry and catalytic mechanism when SACs are applied in Li–O₂ batteries.

With this in mind, to the best of our knowledge, for the first time, we propose a strategy of using a single-atom material as a cathode catalyst for Li–O₂ batteries and develop a strategy inspired by the growth process of strawberries by combining polymer encapsulation with template replication to fabricate a hollow N-doped carbon sphere with isolated single Co sites (N-HP-Co SACs). The atomically dispersed cobalt sites are used to support the ORR and OER reactions. The hollow substrate is used to support the fast mass transfer of electrons, electrolytes, and O₂. During the discharge process, the homogeneous atomic sites serve as nucleation sites for the growth of isolated nanosheet-like $Li_2O_2$, thus favouring subsequent decomposition during charging. More $LiO_2$ intermediates form in the electrolyte by the SACs during charging, which can promote $Li_2O_2$ decomposition through a one-electron process. Benefiting from these merits, Li–O₂ batteries with these N-HP-Co SACs exhibit relatively low overpotential, high specific capacity, and superior rate capability. Moreover, their by-products can be effectively consumed with the help of SACs while displaying excellent cycling stability.

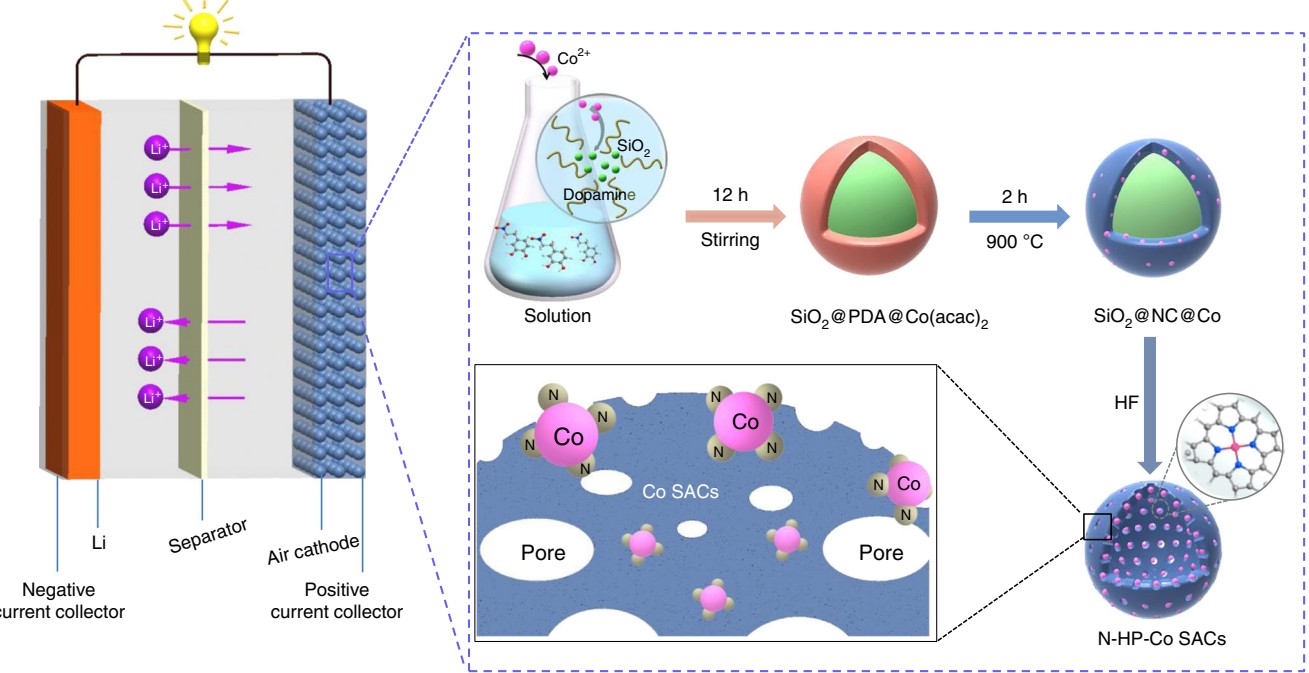

**Fig. 1 Schematic illustration showing the synthesis of N-HP-Co SACs.** The composition in the cell (left) and the synthesis procedures for the N-HP-Co SACs (right).

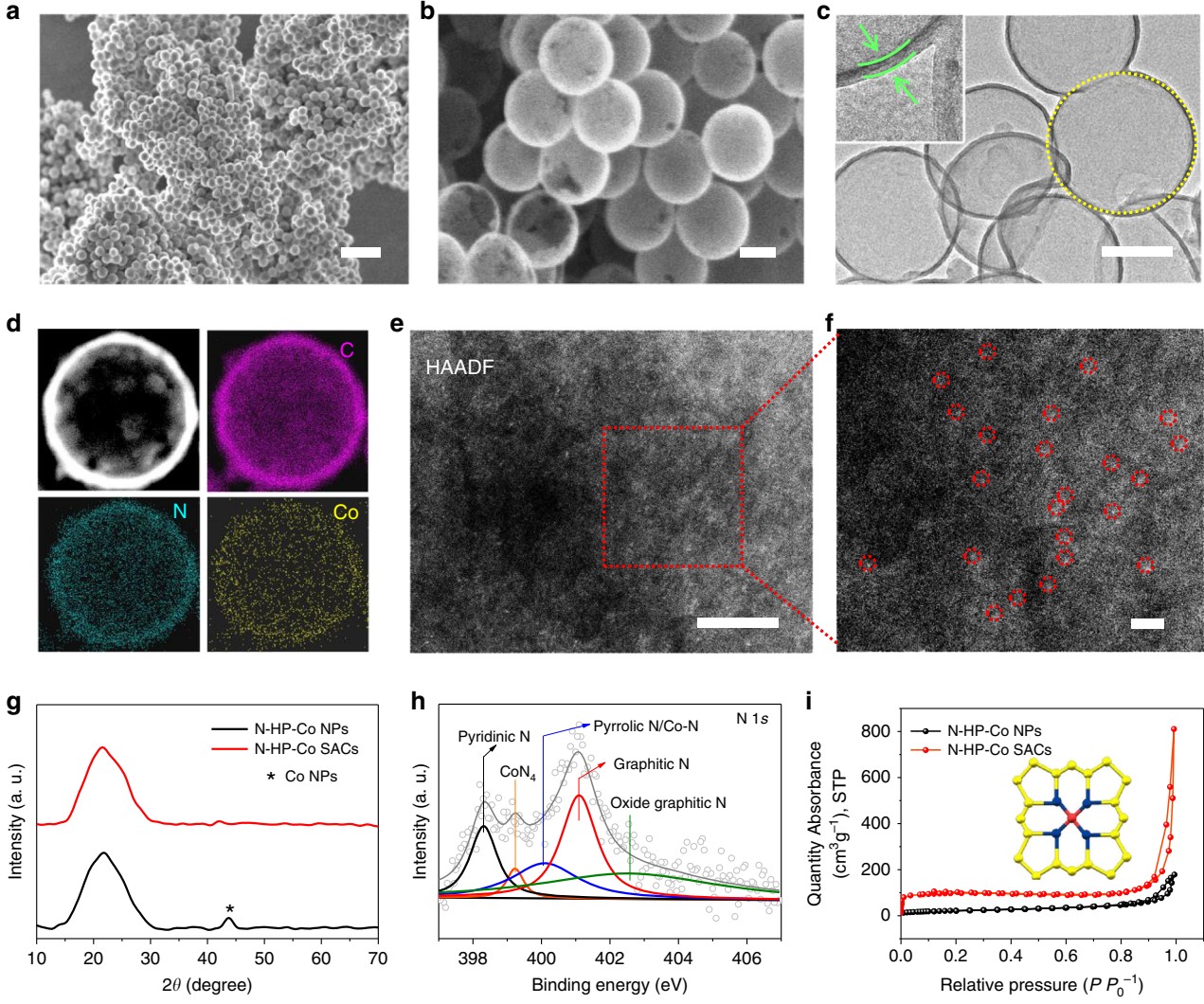

**Fig. 2 Characterization of N-HP-Co SACs.** SEM images of N-HP-Co SACs, 1 μm in (**a**), 200 nm in (**b**); **c** TEM image of N-HP-Co SACs. **d** HAADF-STEM image, 50 nm in (**d**); and the corresponding elemental maps. **e** HAADF-STEM image, 5 nm in (**e**), **f** enlarged image of N-HP-Co SACs, 2 nm in (**f**); **g** PXRD graphs of N-HP-Co NPs and Co-SACs. **h** High-resolution N 1s XPS spectra of N-HP-Co SACs. **i** N$_2$ adsorption-desorption isotherms of N-HP-Co NPs and Co SACs at 77 K. BET: 80.5 cm$^3$ g$^{-1}$ with N-HP-Co NPs; 312.3 cm$^3$ g$^{-1}$ with N-HP-Co SACs.

## Results

**Synthesis and structure of N-HP-Co SACs.** The synthesis procedures for the hollow N-doped porous carbon sphere-structured atomically dispersed Co-N-C catalysts are depicted in Fig. 1. Inspired by an ingenious growth process in nature, we herein demonstrate the "aggregate fruit" electrocatalyst design for Li–O$_2$ batteries. First, a spherical silica template with a particle size of ~400 nm (Supplementary Fig. 1a) was prepared by the Stöber method[30]. The cobalt complexes are mixed with dopamine monomers, which polymerize on the silica spheres to form SiO$_2$@PDA@Co(acac)$_2$ nanospheres (Supplementary Fig. 1b). Then, SiO$_2$@N-doped carbon@Co porous carbon spheres are obtained by pyrolysis to convert the coated poly-dopamine into carbon, followed by etching the silica core with HF; thus, nitrogen-hollow-porous-Co single-atom catalysts (N-HP-Co SACs) can be successfully obtained. During the polymerization process, the single Co atom is the "seed", which grows into the fruitlet, while the carbon is the "fruit", thus forming the "aggregate fruit". This special structure provides more stable Co SACs owing to strong metal-support interactions.

Scanning electron microscopy (SEM) and transmission electron microscopy (TEM) imaging show that the N-HP-Co SAC samples consist of nanospheres with uniform diameters of ~400 nm, and the coating thickness is ~10 nm (Fig. 2a–c). However, the existence and atomic distribution of Co elements are identified via the high-angle annular dark field scanning transmission electron microscopy (HAADF-STEM) image and corresponding EDX spectra for the composite materials. Figure 2d demonstrates that C and N are homogeneously distributed on the carbon, while no Co nanoparticles (Co NPs) are observed. Numerous uniformly distributed white bright dots at the atomic scale are detected on the surface of the carbon spheres, which are assigned to the Co atoms (Fig. 2e, f). In contrast, the image of Co NPs with a size of ~10 nm can be clearly observed through TEM (Supplementary Fig. 2). The Co content of the sample is found to be 0.6 wt% by the EDX spectra (Supplementary Fig. 3), which is nearly in accordance with the value of 0.56 wt% by inductively coupled plasma optical emission spectroscopy (Supplementary Table 1). As presented in Fig. 1g, the X-ray diffraction (XRD) pattern of the N-HP-Co NPs displays a sharp peak that can be indexed to the crystal surface of metallic cobalt. No other peaks related to metallic Co or other cobalt compounds are observed in the N-HP-Co sample, further demonstrating the single-atom nature of the Co. Notably, all the

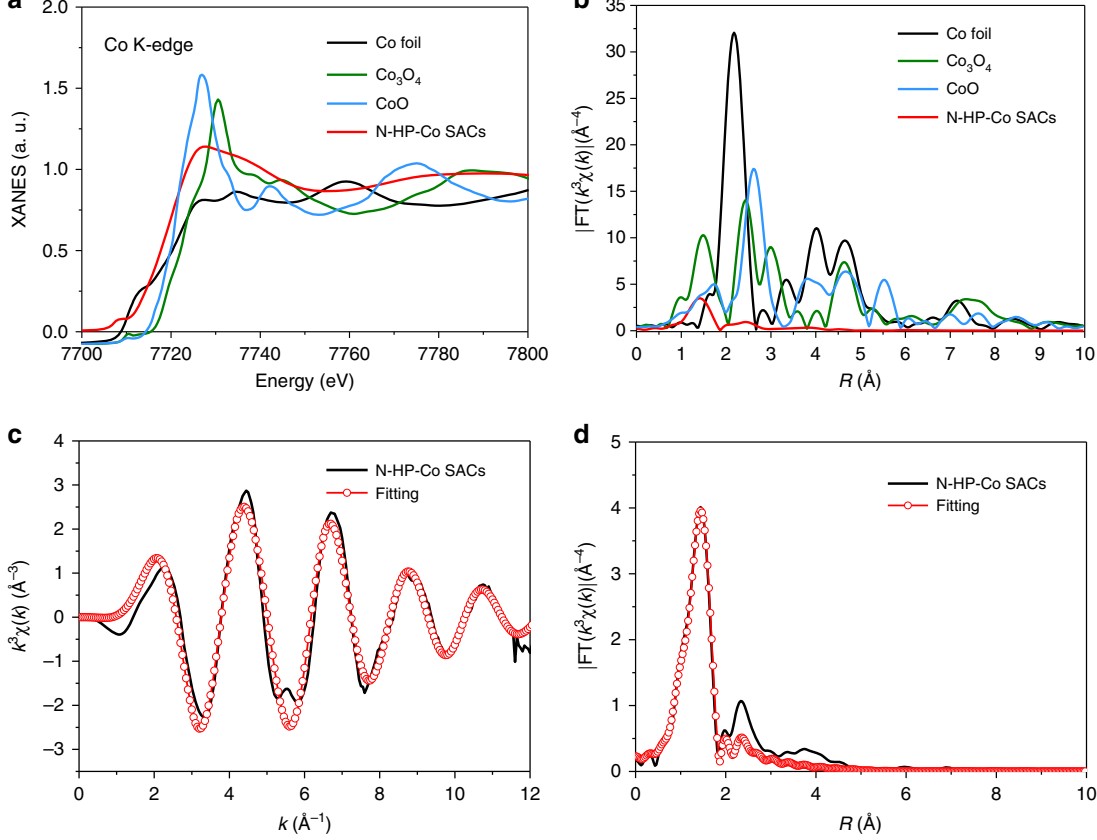

**Fig. 3 Atomic structural analysis. a** XANES spectra and **b** Fourier transform of the Co K-edge of N-HP-Co SACs Co$_3$O$_4$, CoO, and Co foil. **c**, **d** The corresponding EXAFS fitting curves of N-HP-Co SACs at the $k$ space and $R$ space, respectively.

samples exhibit similar carbon structures with dominant D and G bands at 1350 and 1585 cm$^{-1}$ (Supplementary Fig. 4), which are associated with the disordered carbons and $sp^2$ hybridized graphitic carbons, respectively[31]. Remarkably, the N-HP-Co SACs demonstrate a relatively high $I_D/I_G$ value of 0.91, suggesting that they have the largest number of defects caused by the Co atoms in the carbon spheres, which can act as active sites for subsequent reactions. X-ray photoelectron spectroscopy (XPS) was conducted to investigate the binding states of Co, N and C in the N-HP-Co SAC catalysts, as shown in Fig. 2h. XPS presents the peaks assigned to Co, N, O and C in the survey spectrum of the N-HP-Co SACs, illustrating the existence of Co and N in the graphitic carbon framework (Supplementary Fig. 5a). The high-resolution XPS spectrum of N 1$s$ can be divided into five peaks, which correspond to pyridinic-N, CoN$_4$, pyrrolic-N/Co–N, graphitic-N and oxide graphitic-N. In addition, pyridinic-N provides coordination sites to atomic Co in the form of Co–N, and graphitic-N affects the geometric and electronic structure of the carbon skeleton; graphitic-N groups also reveal a positive function in improving the limiting current density of the catalyst towards ORR[32]. Furthermore, the peak at 285.6 eV, which is assigned to the carbon atom bonding with N in the C 1$s$ spectrum (Supplementary Fig. 5b), further confirms the formation of N-doped graphitic carbon. Specifically, compared with the content of $sp^2$ C, the as-prepared N-HP-Co SACs contain much less $sp^3$ C, indicating enhanced electro-conductivity[33]. The above characteristics indicate that in the N-HP-Co SACs, Co is atomically dispersed in the N-doped hollow porous carbon matrix, and a Co atom may coordinate with four N atoms around it, i.e., the formation of a CoN$_4$ moiety. Importantly, the N-HP-Co SACs possess a large BET-specific

surface area of 312.3 cm$^3$ g$^{-1}$ and pore volume of 1.3 cm$^3$ g$^{-1}$ (Fig. 2i, Supplementary Fig. 6), which provide channels for fast mass transport of oxygen and electrolytes and exposure of the abundant active sites.

As shown in Fig. 3, the local chemical bonding state of the Co atoms in the N-HP-Co SACs was further verified by X-ray absorption fine structure (XAFS) measurements. The Co K-edge absorption near-edge structure (XANES) spectra of the N-HP-Co SACs were obtained, as well as those of Co foil, Co$_3$O$_4$, and CoO to be used for comparison (Fig. 3a). In detail, the near-edge absorption energy of the N-HP-Co SACs is suited between that of the Co foil and CoO, suggesting that the isolated Co atoms bear a positive charge between the metal cobalt with zerovalent Co (0) and CoO with bivalent Co (II); the above result indicates the N-coordinated chemical state of the single Co atoms. Compared with the Co foil, the Fourier transform (FT) k$^3$-weighted extended X-ray absorption fine structure (EXAFS) spectrum of the N-HP-Co SACs shows no appreciable Co–Co coordination peak or other high shell peaks. As expected, the FT EXAFS spectrum of N-HP-Co SACs displays a strong peak at 1.4 Å, demonstrating the main peak belonging to the Co–N scattering path (Fig. 3b). Furthermore, the coordination number of the centre Co atom is ~4 according to the EXAFS fitting (Fig. 3c, d, Supplementary Fig. 7), and the mean Co–N/C bond length is 1.96 Å (Supplementary Table 2), further indicating the formation of a CoN$_4$ moiety.

**Discharge behaviour with the N-HP-Co SAC catalysts.** The electrochemical behaviour evaluated in the Li–O$_2$ batteries is shown in Fig. 4. For comparison, a 0.6% commercial Pt/C was

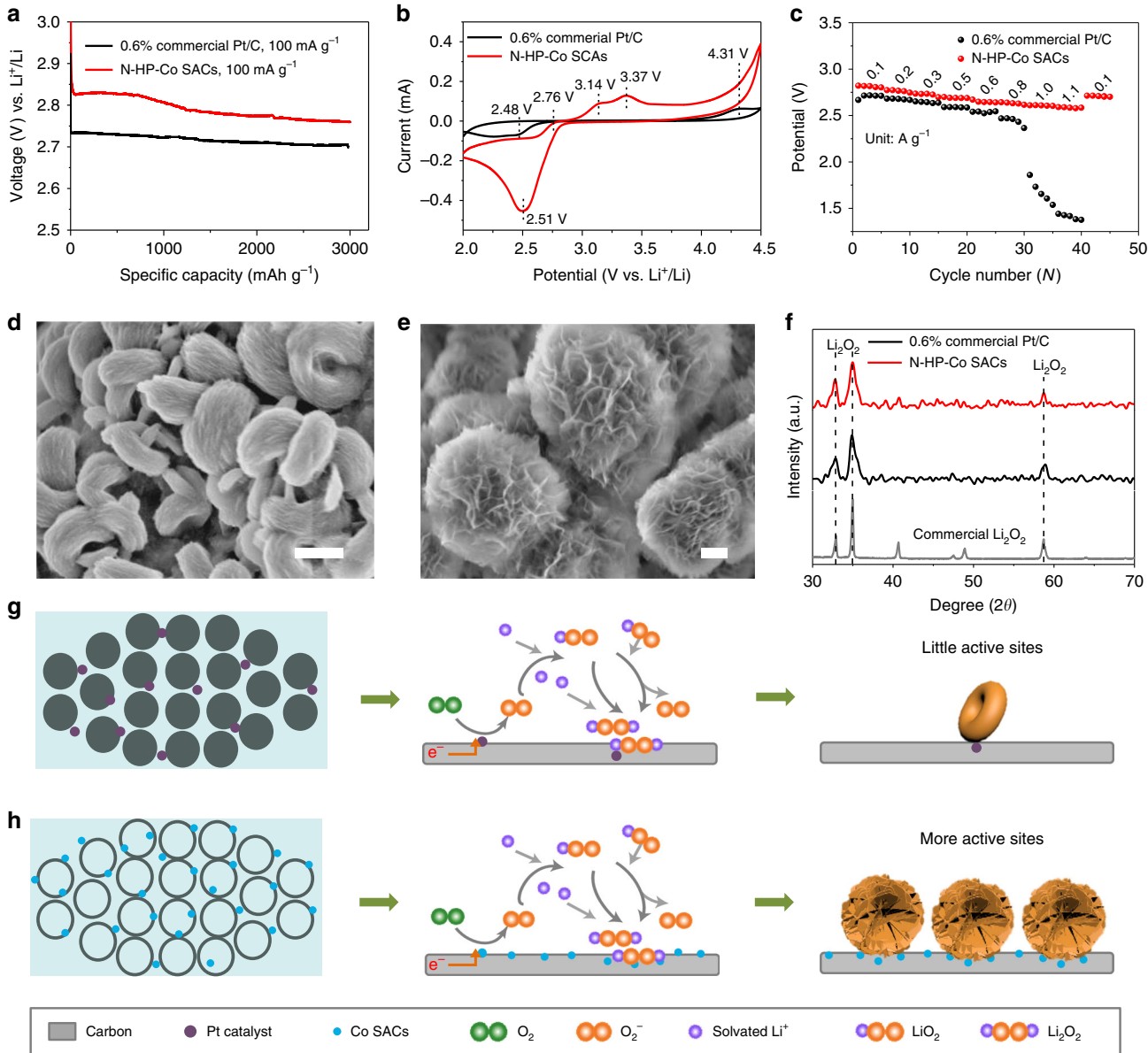

**Fig. 4 Discharge performance and characterization of discharge products. a** First discharge curves with 0.6% commercial Pt/C and N-HP-Co SACs at a current density of 100 mA g$^{-1}$ with a limiting specific capacity of 3000 mAh g$^{-1}$. **b** CV curves. **c** Full range rate performances of Li-O$_2$ batteries at different current densities. **d**, **e** FESEM images of the discharged cathodes with (**d**), 0.6% commercial Pt/C and (**e**) N-HP-Co SACs at a current density of 100 mA g$^{-1}$ with a limiting specific capacity of 3000 mAh g$^{-1}$, 500 nm in (**d**, **e**); **f** The corresponding XRD patterns. The standard spectra for Li$_2$O$_2$ are also shown for reference. Discharge mechanism of 0.6% commercial Pt/C (**g**) and N-HP Co SACs (**h**) in the Li-O$_2$ batteries.

also measured under the same conditions. Figure 4a shows that a low discharge voltage (0.145 V) can be achieved with the N-HP-Co SACs. In addition, the ORR polarization curves (Supplementary Fig. 8) suggest that the kinetics of Li$_2$O$_2$ deposition are greatly improved by the special SACs, which is further supported by the CV curves in Fig. 3b. Compared with 0.6% commercial Pt/C catalyst-loaded cathode, a higher onset potential and larger peak current density are obtained with N-HP-Co SAC catalysts for ORR, thus suggesting the enhanced activity on ORR with SACs. The superior ORR activity of the N-HP-Co SAC catalysts can be attributed to the following two reasons: first, the charge redistribution induced by the decoration of isolated Co atoms improves the O$_2$ adsorption and reduction efficiency and second, the abundance of exposed active centres are provided by the special architecture, which guarantees facile electrocatalytic kinetics[34].

The superior ORR activity of N-HP-Co SACs further inspired us to explore the rate performance. As expected, the rate performance investigations (Fig. 4c) show that the discharge voltage plateau of the N-HP-Co SAC catalyst is higher than that of 0.6% commercial Pt/C at each current density. The first full discharge curves (Supplementary Fig. 9) of the two catalysts and the other commonly used catalysts in Li–O$_2$ cells, along with the EIS results (Supplementary Fig. 10), are consistent with the above result, which indicates more efficient electron, ionic and mass transport. Such a large difference is attributed to the large number of exposed CoN$_4$ sites on the surface of the N-HP-Co SACs, which is beneficial for absorbing oxygen molecules and promoting electron transfer.

Moreover, the morphology of Li$_2$O$_2$ can also be manipulated in situ by virtue of the surface properties of the Co SACs. Figure 4d, e show the SEM images of discharged cathodes with

the 0.6% commercial Pt/C and N-HP-Co SAC catalysts. Notably, the morphology of the discharge products is dramatically different even with the same discharge current density of 100 mA g$^{-1}$. On the cathodes with the 0.6% commercial Pt/C catalyst, the discharge products demonstrate a toroidal morphology on the carbon surface after discharge, which is consistent with the results obtained by other groups[35,36]. However, when the N-HP-Co SACs was used as the cathode, unique nanosheets uniformly grow on the wall of the porous carbon spheres. Such observations also directly indicate the strong interaction of the metal atoms with the supports and the dynamic structural transformation during the discharge process[37]. The PXRD patterns, FTIR and XPS spectra (Fig. 4f, Supplementary Figs. 11 and 12) for the carbon cathode after the discharge process also show that the main discharge product is $Li_2O_2$. Clearly, the nanosheet growth corresponds to the complex electronic structure at the $CoN_4$ sites where the $Li^+$, electrons and $O_2$ meet and are subsequently fed to the surface as "seeds" and then to the bottom side as $Li_2O_2$. This indicates that the growth process is sustained by continuous mass transport to the top of the nanosheets, which is benefited by the directional alignment of the Co SACs. The functional schematic illustration of the catalysts in Li–$O_2$ batteries is shown in Fig. 4g, h. The increased discharge capacity of the N-HP-Co SAC is due to the increased $CoN_4$ as "active seeds" in the N-doped hollow carbon matrix. The $Li_2O_2$ nanoparticles are preferentially formed on the cathode, which can serve as nucleation sites for subsequent $Li_2O_2$ growth. Under the same current and capacity, the N-HP-Co SAC will provide more nucleation sites that are favourable for the formation of small nanosheets and the further self-assembly of large sheets according to the enhanced solvation-mediated mechanism[36]. Furthermore, a special discharge mechanism is in favour of the charge process, such as charge-transport limitations and the electronically insulating property of the discharge products.

**Recharging behaviour with the N-HP-Co SAC catalysts.** For the charging behaviour, Fig. 5 indicates that the charge voltage is much lower than that of the 0.6% commercial Pt/C catalyst loaded on a CP cathode at ~500 mV. To determine whether the improved electrochemical performance is caused by the Co SAC catalysts, the full discharge/charge analysis with the commonly used 0.6% catalyst is displayed in Supplementary Fig. 9. The single Co atom exhibits a lower OER overpotential compared with those of other metal catalysts in Li–$O_2$ batteries (Supplementary Table 4). Moreover, a 5% commercial Pt/C was used for comparison (Supplementary Fig. 9a), and the N-HP-Co SACs exhibit comparable advantages in both the discharge and charge reactions. This fact is further verified by the CV curves (Fig. 4b); three obvious peaks occur during charging for the decomposition of $Li_2O_2$. First, the peak at ~2.76 V may be related to $LiO_2$ oxidation in the electrolyte[38]. The peak at ~3.2 V in the anodic scan of the Co SAC catalysts originates from the suitable decomposition of nonstoichiometric $Li_{2-x}O_2$, especially outside of $Li_2O_2$. The third peak at ~3.3 V can be assigned to the oxidation of $Li_2O_2$, which is mainly formed by the solvation-mediated discharge process through a one-electron transfer. However, only one broad weak peak at 4.31 V is observed in the Pt/C electrode, thus demonstrating fewer chances for the dissociation of the formed $Li_2O_2$ followed by a two-electron transfer or other reaction products such as LiOH or $Li_2CO_3$ requiring high thermodynamic potentials[39]. These results indicate that the N-HP-Co SACs possess excellent functional activities towards the ORR/OER, which are beneficial for improving the Li–$O_2$ battery performance. For the charging process, the peaks of $Li_2O_2$ disappear, and the XRD patterns (Supplementary Fig. 13) are the same as

those of the pristine cathode before discharge, which is consistent with the impedance analysis results (Supplementary Fig. 10). It is generally accepted that the electrochemical decomposition of $Li_2O_2$ will follow these two steps: first, $Li_2O_2 \rightarrow Li^+ + e^- + LiO_2$, and second, $LiO_2 \rightarrow Li^+ + e^- + O_2$. Simply, $Li_2O_2$ is first extracted from a Li atom to form a $LiO_2$ intermediate. The $LiO_2$ consequently releases oxygen through electrochemical or chemical reactions in the presence of effective catalysts. To verify the $Li_2O_2$ oxidation on the N-HP-Co SACs, UV–vis measurements were used to quantitatively monitor the $LiO_2$ intermediates[40,41]. Based on the spectra obtained from the DMSO solution dissolved with a certain amount of $KO_2$ (Supplementary Fig. 14), the absorption peak at 262 nm can be assigned to superoxide species. As shown in Fig. 5a, the spectra of the different charged electrodes have a similar absorption peak of the superoxide intermediates at ~262 nm, which confirms the two-step decomposition of $Li_2O_2$. More significantly, the UV–vis absorbance shows that the two types of cathodes have significant differences even at the same charge stage. Obviously, the absorbance values from the N-HP-Co SACs were all higher than those of the commercial Pt/C catalyst at stages a and b (the state in Fig. 5a), which means that more $LiO_2$ intermediates form during the charging process. Thus, the extraordinary SACs can bridge the gap between homogeneous and heterogeneous catalysis by virtue of the complex coordination environment around the active centres, quantum size effect and support effect[42]. Compared with heterogeneous catalysts, Co SACs exhibit a low metal loading but high atom utilization and homogeneous active $CoN_4$ sites with tunable electronic environments towards $Li_2O_2$ oxidation for high catalytic activity through a one-electron reaction. As shown in Supplementary Figs. 15 and 16, more $O_2^-$ existed in the electrolyte with N-HP-Co SACs according to the nitrotetrazolium blue chloride detection and EPR spectra[40,43].

Based on the above results, the charging mechanism is presumed in Fig. 5b. At the relatively low charge potential of the Co SACs (Fig. 5a, Supplementary Fig. 9), the nanosheet-like $Li_2O_2$ may be the first to undergo a topotactic delithiation to form off-stoichiometric $Li_{2-x}O_2$. The N-doped $sp^2$ carbon can improve the binding energies, further regulating the electron structure and the adsorption energy of the intermediate $LiO_2$. Then, the increased $LiO_2$ intermediates indicate that the recharge process will probably convert from a two-electron reaction to a single-electron reaction. The main postulation based on the complex environment around single $CoN_4$ sites is that they will interact preferentially with $LiO_2$ based on the ever-changing electron densities, further modulating the electrochemical activity during charging. Density functional theory (DFT) calculations were further used to understand the possible effect of the cathode surface properties on the $Li_2O_2$ decomposition pathway. The optimized structures and the corresponding binding energies between the reduced species ($LiO_2$ ion pairs) on the Pt/C ((111), according to Supplementary Fig. 17), N-HP-CS, and N-HP-Co SAC surfaces are shown in Fig. 5c, d, f, and g and Supplementary Fig. 18. We find that the binding energies of the reduced species on the Pt/C (−1.85 eV) and N-HP-CSs (−1.55 eV) surfaces are all much higher than those on the N-HP-Co SAC surface (−1.03 eV), indicating that the "agravic" species more easily spread to the electrolyte from the N-HP-Co SAC surface. Therefore, the decomposition of $Li_2O_2$ is likely to be a one-electron reaction, which is in good agreement with the experimental observations (Fig. 5a, Supplementary Figs. 15 and 16). Moreover, the corresponding charge density difference diagrams of the Pt (111) and $CoN_4$ systems are shown in Fig. 5e, h; the yellow area is the gained electron and the blue area is the lost electron, indicating that the reaction occurs on the $CoN_4$ sites with an abundance of electron transfers compared with that on

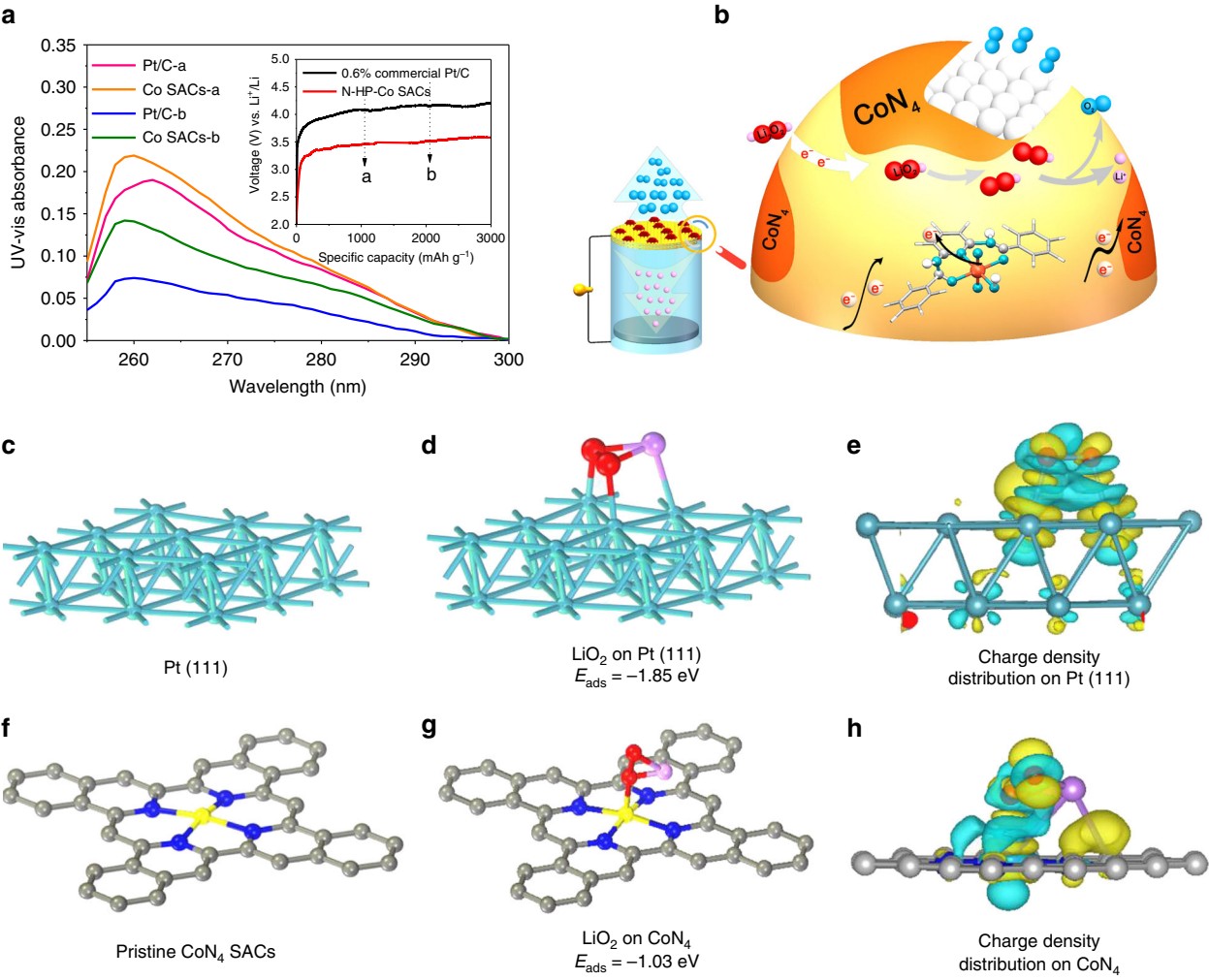

**Fig. 5 Charge characterization. a** UV-vis spectra of DMSO-extracted superoxide products from the 0.6% commercial Pt/C and N-HP-Co SAC cathodes at different charging states (**a** state a, charge to 1000 mAh g$^{-1}$; **b** state b, charge to 2000 mAh g$^{-1}$). The inset of **a** shows the corresponding charge potential profiles of the Li−O$_2$ cells using 0.6% commercial Pt/C and N-HP-Co SACs at a current density of 100 mA g$^{-1}$. **b** Schematic illustration showing the charge mechanisms of the N-HP-Co SAC-catalysed Li-O$_2$ batteries. **c**, **d** Pristine and top views of the optimized structures with the corresponding binding energy of LiO$_2$ on Pt (111); **f**, **g** optimized structure and the corresponding binding energy of LiO$_2$ on CoN$_4$; and **e**, **h** corresponding charge density distribution. Colour code: platinum (green), carbon (grey), CoN$_4$ (blue), and LiO$_2$ (red).

the Pt sites. These results reveal that the single Co atom in the Li–O$_2$ batteries can form a more positive equilibrium system for the formation and consumption of Li$_2$O$_2$. However, we cannot ignore the effect of the discharge products on the OER because the shape and size of Li$_2$O$_2$ can significantly influence the charge potential profiles, which are further demonstrated by "charge-only" reactions (see Supplementary Figs. 19, 20 and 21).

**Stability of the electrochemistry in Li–O$_2$ batteries**. We further examined the stability of the above reaction system with Co SACs. Figure 6a shows that the N-HP-Co SAC-loaded electrode exhibits stable discharge/recharge reactions for more than 250 cycles (218 days), whereas the 0.6% commercial Pt/C electrode degrades after only 100 cycles when the terminal voltage reaches 2 V. More importantly, the Li–O$_2$ cells with the N-HP-Co SACs still exhibit a good, stable specific capacity and retain a relatively stable terminal discharge voltage above 2 V after ~50 cycles at a large current density of 400 mA g$^{-1}$, indicating excellent cycling stability. Figure 6b, d shows the morphology of the discharged carbon electrodes after the 5th and 20th cycles. In the presence of the 0.6% commercial Pt/C catalyst, the particles with the pristine

toroidal morphology disappear instead of the film-like discharge products. However, with the N-HP-Co SACs, only nanosheets, in a clear and compact manner, can be observed on the cathode surface after the 20th cycle (Fig. 6c, e). Importantly, the PXRD patterns (Supplementary Fig. 23) match well with the above results.

The degradation of the crystalline Li$_2$O$_2$ in cycling can be speculated to be the increasing accumulation of the side products on the surface of the cathode, which goes against the nucleation and crystallization of Li$_2$O$_2$ during the subsequent discharge, and leads to the formation of amorphous Li$_2$O$_2$. Figure 6f, g and Supplementary Fig. 24 show the Li 1$s$, O 1$s$ and C 1$s$ spectra of the discharged cathodes after consecutive cycles. The Li 1$s$ and O 1$s$ peaks of Li$_2$O$_2$ at 55.04 and 531.62 eV, respectively, are consistent with previously reported XPS studies on Li$_2$O$_2$ formed in Li–O$_2$ cells[44,45]. However, as the cycle continues, the intensity obviously decreases, and a peak of Li$_2$CO$_3$ begins to appear on the cathodes of the 0.6% commercial Pt/C catalyst. This result corresponds well with the changes in the C 1$s$ peaks (Supplementary Fig. 24); in detail, the C 1$s$ spectra of 0.6% commercial Pt/C catalyst confirms the formation of a variety of parasitic C-containing products. For instance, the peaks at 288.7

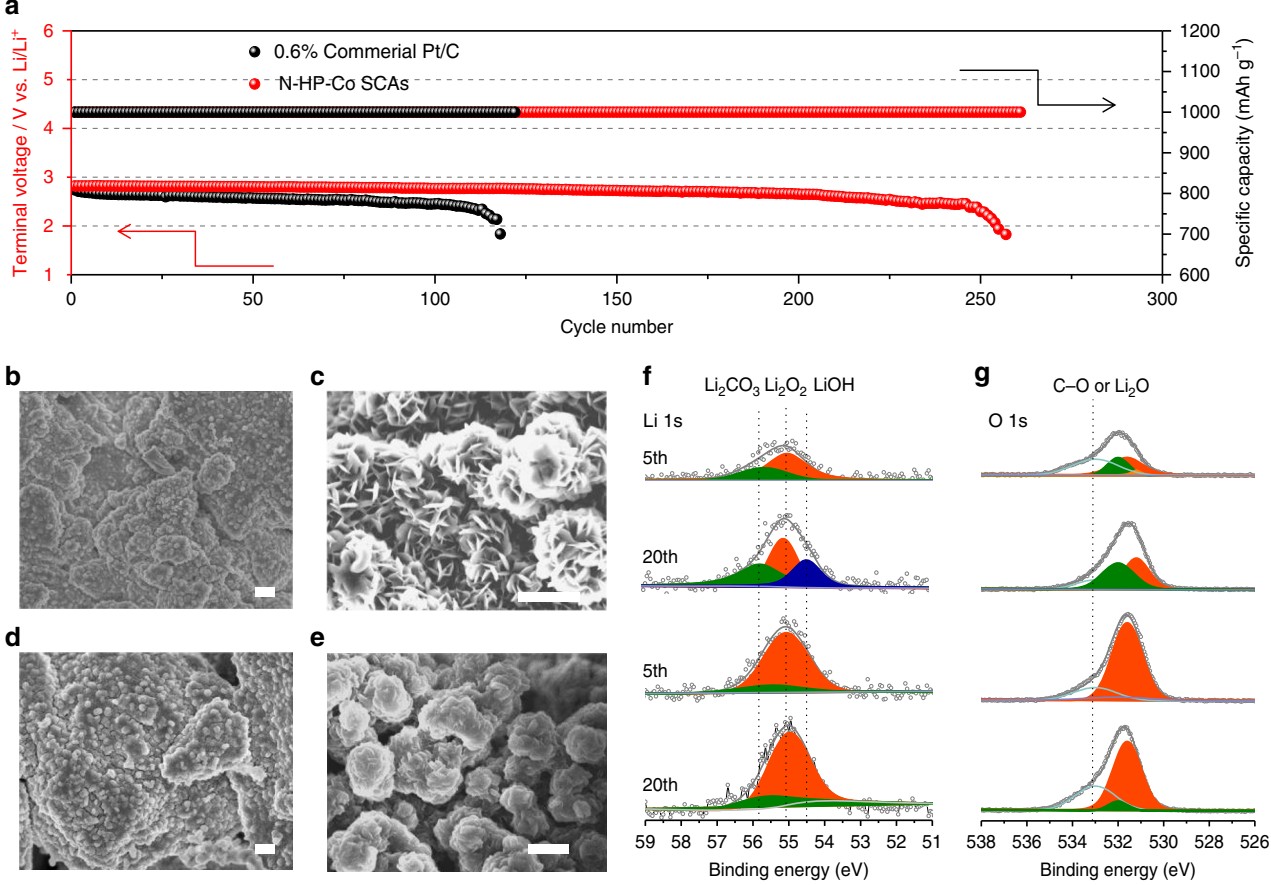

**Fig. 6 Cycling stability of Li–O₂ batteries with N-HP-Co SACs. a** Voltage versus cycle number on the discharge terminal of the Li–O₂ cells with commercial Pt/C and N-HP-Co SACs. FESEM images of the discharged cathode with the 0.6% commercial Pt/C catalyst (**b**) and the N-HP-Co SAC catalyst (**c**) at a current density of 200 mA g⁻¹ after the 5th cycle and the cathode with the 0.6% commercial Pt/C catalyst (**d**) and the N-HP-Co SAC catalyst (**e**) after the 20th cycle. One micrometer in **b**, **d**; 500 nm in **c**, **e**; **f**, **g** XPS spectra of the discharged cathodes with 0.6% commercial Pt/C and N-HP-Co SACs: Li 1s (**f**) and O 1s (**g**) after the 5th and 20th cycles.

and 289.89 eV exhibit the formation of a small amount of lithium carboxylates (ROCO₂Li) and carbonates (Li₂CO₃) upon the first discharge (Supplementary Fig. 12c). However, the amounts of these products increase with further cycling. In sharp contrast, minor peaks referring to by-products are found with the N-HP-Co SAC-containing cathodes after the 20th cycle because of potential side reactions that occur between the discharge products and electrolyte solvent. These results are also confirmed by an FTIR analysis (Supplementary Fig. 25). Furthermore, the side products during the charging process were monitored by FTIR and NMR measurements, and the results are shown in Supplementary Figs. 26 and 27.

Based on the above results, the gas evolution during the charging reaction was monitored through an in situ differential electrochemical mass spectrometry analysis. As shown in Supplementary Fig. 28, the curves of O₂ indicate that the main reaction is Li₂O₂ decomposition. Compared with N-HP-Co SACs, the CO₂ pressure for 0.6% commercial Pt/C is very large and increases with further cycling, which is possibly due to the side reactions on the carbon or the binder by a highly reactive radical. In addition, gas chromatography signals (Supplementary Fig. 29, Supplementary Table 3) were used to quantitatively analyze the reversibility of the reactions and indicated the O₂ amount (1.16 μL) of the N-HP-Co SACs after the first recharge. In addition, the difference can be illustrated by the TiOSO₄-based quantitative results (Supplementary Fig. 30), in which the red bars are caused by the incomplete decomposition of Li₂O₂[46]. Specifically, the cell

with the N-HP-Co SACs exhibits both a high Li₂O₂ formation efficiency (93.9%) and low Li₂O₂ residual (4.8%); specifically, the difference becomes increasingly clear with further cycling.

**Durability of the N-HP-Co SACs after cycling in Li–O₂ batteries.** In this regard, we conclude that the high surface energy of the Co SACs can protect the carbon support from the attack of the O²⁻ ion, thereby reducing the parasitic reactions on the carbon or binder. As shown in Fig. 7a, the XPS of the Co 2p spectrum of the recharged cathode demonstrates that single CoN₄ active sites still exist, and no Co–Co is observed. Because of the difficulty in observing the carbon support surface after 50 cycles, a relatively uniform distribution of Co and N is observed through EDX elemental mapping (Fig. 7b); the XAFS measurements (Fig. 7c, d) further demonstrate the single Co atom in the N-doped carbon sphere support. This indicates that Co atoms are still bound to the carbon surface through a nitrogen-related ligand. However, as the cycling proceeds, the content of Co is unavoidably reduced. This is the next important problem that we need to overcome. In summary, the N-HP-Co SACs exhibit excellent activity, durability and stability in Li–O₂ batteries.

## Discussions

In summary, inspired by the growth process of strawberries, a well-dispersed single Co atom catalyst was first prepared by a polymer encapsulation strategy with a SiO₂ template and then

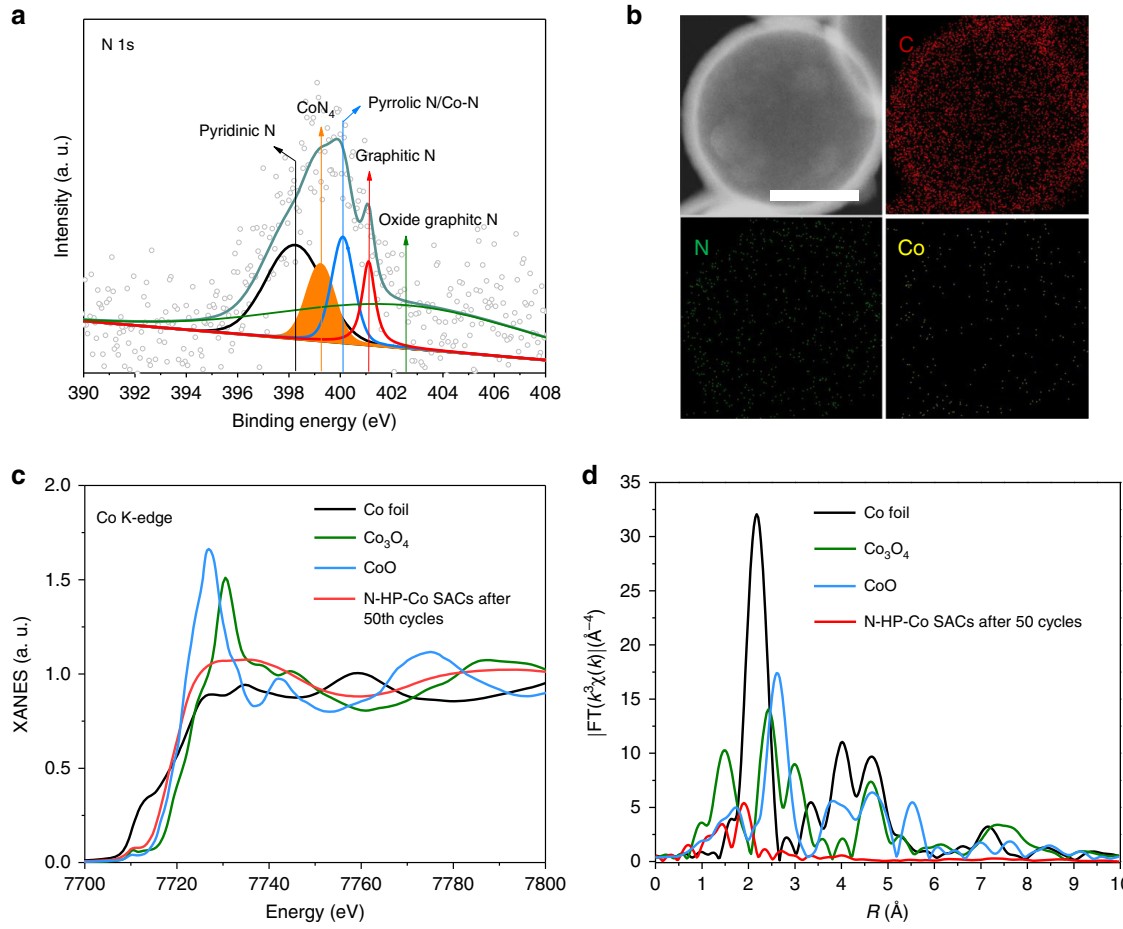

**Fig. 7 Cycling stability of N-HP-Co SACs. a** N 1s XPS spectra for N-HP-Co SACs after 50 galvanostatic cycles. **b** HAADF-STEM images and the corresponding EDX elemental maps after 50 galvanostatic cycles, 200 nm in **b**. **c**, **d** XANES spectra and Fourier transform of the Co K-edge of N-HP-Co SACs, $Co_3O_4$, CoO, and Co foil.

pyrolysis under an inert atmosphere. The single-atom nature of the active centres in N-HP-Co SACs and the resulting low-coordination environments and enhanced metal-support interactions provided remarkable catalytic performance and successfully regulated the deposition/decomposition routes in Li–O₂ batteries. Compared with commercial Pt/C catalysts, the highly reactive activity Co SACs could attack more $O^{2-}$, which not only accurately manipulated the discharge product but also increased the discharge capacity, thus contributing greatly to alleviating cathode passivation and side reactions. As a result, the Co SAC-catalysed Li–O₂ batteries showed an ultralong cycle life (261 cycles at a current density of 100 mA g⁻¹ with a cut off capacity of 1000 mAh g⁻¹) and a high discharge capacity (~14,777 mAh g⁻¹ at a current density of 100 mA g⁻¹). For the charging process, the initial delithiation of $Li_2O_2$ was a one-electron process rather than the general two-electron mechanism, which was achieved by the dispersed single Co atom catalysts in the Li–O₂ cells. The one-electron process is more kinetically favourable and highly reversible, leading to identical charging behaviour with noble-based catalysts. This work opens a new approach for the rational design of highly efficient noble metal-free electrocatalysts for energy storage and conversion applications.

## Methods

**Li–O₂ cell preparation and electrochemical measurements**. The electrochemical performance of the Li–O₂ cell was tested in a 2025-type coin cell. All of the cells were assembled in a glove box in an Ar atmosphere with a lithium foil anode, a glass fibre separator, an oxygen cathode and an electrolyte containing 1 M LiTFSI in TEGDME. A pristine cathode was prepared by coating a homogenous ink, which

was composed of a mixture of 80 wt% N-HP-Co SACs and 20 wt% poly(vinylidene fluoride), onto a CP current collector. In addition, a mixture of 80 wt% Super P containing 0.6% commercial Pt/C catalyst and 20 wt% poly(vinylidene fluoride) was deposited on CP and prepared in the same manner. The active material loading was ~0.35 mg cm⁻². The electrochemical performances of the cathodes with N-HP-Co SAC and commercial Pt/C catalysts were tested in a specific capacity-controlled mode under various current densities. The electrochemical impedance spectroscopy of the cell was evaluated using a CHI660E electrochemical workstation (Shanghai chenhua instrument co., Ltd) in a frequency range of 10⁵–10⁻² Hz.

**Synthesis of N-HP-Co SACs by a polymer encapsulation approach**. Silica spheres were used as templates to prepare hollow carbon spheres. In a typical experiment, $SiO_2$ (0.15 g), dopamine (0.15 g) and cobalt acetylacetonate (3.6 mg) were dissolved in 100 mL deionized water and stirred for 10 min. Tris-buffer (1.21 g) was added to adjust the pH of the resulting solvent to 8.5 and stirred for 12 h. The resulting Co(acac)₂@SiO₂@PDA nanospheres were suction filtrated and washed with distilled water several times. The N-HP-Co SACs were obtained by pyrolyzing Co(acac)₂@SiO₂@PDA at 900 °C in a N₂ atmosphere and etching the silica core with HF.

**Characterizations**. The morphology and structure of the materials were characterized using various physicochemical techniques, including XRD, HAADF-STEM, XAFS measurements, field emission scanning electron microscopy (FESEM), TEM and XPS. The discharge and recharge products were characterized using XRD, SEM, NMR, FTIR, XPS, and UV–vis absorption spectrum technology. HAADF-STEM images and corresponding EDX elemental maps were characterized by using a JEM-ARM300F Grand ARM atomic resolution electron microscope with double Cs correctors. TEM was carried out with a JEM-2100F transmission electron microscope.

**Theoretical calculations**. All electronic structure and energy calculations were performed by spin-polarized DFT using the Vienna ab initio simulation package[47–50].

The projector-augmented wave (PAW) potentials were used to describe ion core and valence electron interactions[51,52]. A generalized gradient approximation with the Perdew–Burke–Ernzerhof functional[53] was selected to describe the exchange-correlation functional. A kinetic energy cut off of 400 eV was used with a plane-wave basis set. The integration of the Brillouin zone was conducted using a $3 \times 3 \times 1$ Monkhorst-Pack grid[54]. The convergences of the force and the total energy were set as 0.05 eV Å$^{-1}$ and $1.0 \times 10^{-4}$ eV atom$^{-1}$, respectively. Van der Waals (VDW) interaction was employed in our calculations by the Rugers-Chalmers Van der Waals Density Functional (VDW-DF) approach[55,56]. The Pt (111) surface was obtained by cutting a Pt crystal along the [111] direction, and a $3 \times 3$ unit cell with three layers (Pt$_{36}$) was chosen. The atoms in the bottom two layers were fixed in their bulk positions, and those in the other layers were allowed to relax. The CoN$_4$ and the different types of N-doped carbon supports were prepared by a $5 \times 5$ supercell single layer graphene. A large distance of 15 Å was selected to avoid an imaging interaction. To study the stability of LiO$_2$ on the selected materials, the adsorption energy was defined as the following:

$$E_a = E_{LiO_2} + E_X - E_{LiO_2@X} (X = CoN_4, Pt111, N - doped\,carbon), \quad (1)$$

where $E_{LiO_2@X}, E_X, E_{LiO_2}$ correspond to the total energies of the LiO$_2$ and substrates, an isolated substrate and LiO$_2$, respectively. A positive value suggests strong binding and stable chemisorption.

## Data availability

The data supporting the findings of this study are available within the paper and its Supplementary Information, and from the corresponding author upon reasonable request.

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

## Acknowledgements

This work was financially supported by the National Natural Science Foundation of China (Grant No. 51771177, 51972141), the 111 Project (B17020), the Education Department of Jilin Province (JJKH20190113KJ), Jilin Province Science and Technology Development Program (Grant No. 20190303104SF) and the Jilin Province/Jilin University Co-construction Project-Funds for New Materials (SXGJSF2017-3).

## Author contributions

J.J.X. and L.N.S. developed the concept, designed the experiments. W.Z. and X.G. performed and analyzed the high-angle annular dark field scanning transmission electron microscopy (HAADF-STEM) tests. Y.W was responsible for the theoretical computations. L.C.Z, H.F.W, X.X.W, Q.C.L, and F.L were all involved in discussions. J.J.X., L.N.S., W.Z., and Y.W wrote the paper. All of the authors discussed the results and reviewed the manuscript.

## Competing interests

The authors declare no competing interests.
