## [Peer Review File · Nature Communications]

Reviewers' comments:

Reviewer #1 (Remarks to the Author):

The manuscript NCOMMS-19-33424 by Song et al. discusses performance of Co electrocatalyst encapsulated by polymer in Li-O₂ cells. The study includes variety of characterization methods to characterize, however, a major question regarding true reversibility of the cells is not discussed.

Common electrochemical tests such as galvanostatic cycling or cyclic voltammetry do not provide any conclusive information regarding the quantity (or degree) of formation and decomposition of reversible products. Researchers often assume that if discharge and charge capacities are "balanced" this means reversible batteries. It has however shown in many literatures that oxygen batteries can easily suffer formation of side products formed via parasitic reactions. Therefore, it is crucial to quantify the reversibility of reactions. Some of analytical techniques such DEMS (to quantify the amount of consumed/formed O₂) and in-situ XRD (with some added reference materials to quantify the amount of reactions products) can be useful.

Therefore, without quantifying the main reaction products it is not possible to reveal the performance of the proposed catalyst. Thus, the manuscript in its current format is not suitable for publication.

Reviewer #2 (Remarks to the Author):

The manuscript described a single-atom catalyst in Li-O₂ batteries, reducing the discharge and charge overpotentials and improve the cycling efficiency. The cell performance is quite nice, however, I think, the claims are not well supported by the data and I am not convinced by this work. The current version cannot be accepted from my perspective and for the authors, the following issues need to be solved.

1. The authors claimed they have synthesized a single-atom catalyst, however, it is a bit too much to claim it as single-atom catalyst purely based on HAADF-STEM result. In fact, the PXRD stills shows a bit Co signal in Fig. 2g and Figures in SI, and a higher resolution one is needed. XANES, EXAFS, X-ray absorption spectra data, etc. are necessary to prove the idea of single-atom catalyst. The bond length, angle, charge state, tec. between Co and N need further discussion.

2. As comparisons, the cell performance in this manuscript is among the average in the catalyst in Li-O₂ batteries. It is not fair simply compare this with commercial Pt/C catalyst. The Pt/C is not designed for Li-O₂ batteries and the authors should compare with other catalysts in Li-O₂ batteries to prove the importance and benefits of this single-atom catalyst.

3. The measurements in Q1 should be done on the catalysts after cycling to confirm whether the catalyst has been poisoned or not.

4. The quantification of Li₂O₂ is necessary as well after the 1st, 5th and 50th discharge and charge to measure how much Li₂O₂ has been formed and decomposed during cycling.

5. The PXRD quality is low and the signal and noise ratio should be optimised. For FTIR, it would be good to have the band from 4000 cm⁻¹ to 400 cm⁻¹ to demonstrate whether LiOH and other carbonates exists.

6. How can this catalyst have effect both on discharge and charge? The authors need to clarify the mechanism with more evidence.

7. I do not like studies in Li-O₂ batteries where the discharge capacity claimed to be over 15000 mAh/g, while cycling at a 1000 mAh/g capacity. Why not have the full capacity cycling? I don't understand Fig. 3c, voltage vs. capacity in different rates? As the authors claimed this superior rate performance, then why not cycling at relatively higher rates with a much bigger capacity?

In general, without solid evidence of single-atom catalyst plus the average cell performance, I think it

is too brave to prove the single-atom catalyst idea and the authors have not convinced me the importance of this studies.

Response to Reviewers

Sincerely send our thanks to the editor and all reviewers for the efforts on this manuscript. We greatly appreciate the editor and reviewers giving the chance to revise the current version, especially, the insightful comments and helpful suggestions for further improving the quality of the manuscript. We have responded to all the questions raised by the reviewers point by point, and also revised the manuscript with the revisions marked in Yellow in the revised manuscript. The responses to the reviewers' concerns are addressed as follows.

Reviewer #1

Comment: The manuscript NCOMMS-19-33424 by Song et al. discusses performance of Co electrocatalyst encapsulated by polymer in Li-O₂ cells. The study includes variety of characterization methods to characterize, however, a major question regarding true reversibility of the cells is not discussed.

Common electrochemical tests such as galvanostatic cycling or cyclic voltammetry do not provide any conclusive information regarding the quantity (or degree) of formation and decomposition of reversible products. Researchers often assume that if discharge and charge capacities are “balanced” this means reversible batteries. It has however shown in many literatures that oxygen batteries can easily suffer formation of side products formed via parasitic reactions. Therefore, it is crucial to quantify the reversibility of reactions. Some of analytical techniques such DEMS (to quantify the amount of consumed/formed O₂) and in-situ XRD (with some added reference materials to quantify the amount of reactions products) can be useful.

Therefore, without quantifying the main reaction products it is not possible to reveal the performance of the proposed catalyst. Thus, the manuscript in its current format is not suitable for publication.

Response: Many thanks for your evaluation on the topic of our research. We agree with the reviewer that the “it is crucial to quantify the reversibility of reactions”. According to the

reviewers' valuable suggestion, the *in situ* differential electrochemical mass spectrometry (DEMS) of gaseous oxygen evolution and the Gas chromatography (GC) analyze after the charge upon the 1st, 5th and 50th are carried out to testify the reversibility of the battery. The corresponding results and discussion are displayed in Supplementary Figs. 28, 29, Supplementary Table S3 and the revised manuscript (Lines 18-25, Page 13). The DEMS results indicate that much more O₂ is formed in the cell with N-HP-Co SACs than the 0.6% commercial Pt/C and less other signals (such as CO₂, NO and H₂O) can be observed during the charging process at the end of different cycles, which is in accordance with the original SEM, XPS (Fig. 5b-g, Supplementary Figs. 12 and 24), FTIR results (Supplementary Figs. 11, 25, 26) and ¹H NMR results (Supplementary Fig. 27). Based on the quantitative GC results, the charge efficiency of the N-HP-Co SACs after 1, 5, 50 cycles are 87.4%, 84.6% and 52.9%, respectively, much higher than that with 0.6% commercial Pt/C after 1, 5, 20 cycles of 82.9%, 67.7% and 47.8%.

We apologize for the unavailable in-situ XRD analysis limited by the laboratory's available condition. But the UV-vis spectra tests of discharge-charge cathodes are carried out to identify the components of the discharge-charge products and the reversibility of the battery, which is also a commonly used detection method in this field (*Chem*, 2018, 4, 1345–1358; *Science*, 2018, 361, 777-781; *J. Phys. Chem. C*, 2017, 121, 9657-9661; *Angew. Chem. Int. Ed.*, 2017, 56, 4960–4964). The UV-vis spectra were obtained from the pristine TiOSO₄-based standard solution by adding a certain amount of H₂O₂ and the calibration curve for H₂O₂ in Ti(IV)OSO₄ was detected according to the absorbance of UV-vis curves at 405 nm. Based on these calibration curves, the UV-vis measurements of the discharged and recharged cathodes were carried out to indirectly quantify the amount of consumed/formed O₂ and reactions products. And the corresponding results and discussion are displayed in Supplementary Fig. 30 and the revised manuscript (Lines 1-5, page 14). The results indicate that the cell with the N-HP-Co SACs exhibits a higher Li₂O₂ formation efficiency (93.9%) and lower Li₂O₂ residual rate (4.8%) than the cell with 0.6% commercial Pt/C (81.9% and 23.6%) after the

first discharge and recharge, respectively; specifically, the results of that difference become obvious with the increasing cycles.

Taken together, the **original SEM, XPS, FTIR, ¹H NMR** (Fig. 5b-g, Supplementary Figs. 11, 12, 24, 25, 26, 27) and **newly added DEMS, GC, UV-vis spectra** (Supplementary Figs. 28, 29, 30, Supplementary Table S3) results can qualitatively or quantitatively confirm the higher Li₂O₂ formation/decomposition efficiency of cell with N-HP-Co SACs than the cell with the 0.6% commercial Pt/C, demonstrating the superiority of N-HP-Co SACs in improving the reversibility of the Li-O₂ battery.

Thank you so much again for your helpful comments in improving the quality of our manuscript.

Reviewer #2

General Comment: The manuscript described a single-atom catalyst in Li-O₂ batteries, reducing the discharge and charge overpotentials and improving the cycling efficiency. The cell performance is quite nice, however, I think, the claims are not well supported by the data and I am not convinced by this work. The current version cannot be accepted from my perspective and for the authors; the following issues need to be solved.

Response: Thanks very much for the reviewer's valuable comments and helpful suggestions. We have provided a detailed point-by-point response to each question.

Comment 1: The authors claimed they have synthesized a single-atom catalyst; however, it is a bit too much to claim it as single-atom catalyst purely based on HAADF-STEM result. In fact, the PXRD stills shows a bit Co signal in Fig. 2g and Figures in SI, and a higher resolution one is needed. XANES, EXAFS, X-ray absorption spectra data, etc. are necessary to prove the idea of single-atom catalyst. The bond length, angle, charge state, tec. between Co and N need further discussion.

Response: Many thanks for the referee's valuable suggestion. Indeed, it is not enough to prove the obtained catalyst as a single-atom catalyst purely based on HAADF-STEM results. According to the reviewer's suggestion, the XANES, EXAFS, X-ray absorption spectra data are carried out and analyzed in the revised manuscript. The results and corresponding discussion are displayed in Fig. 2, Supplementary Fig. 7, Supplementary Table S2 and the revised manuscript (Lines 4-18, page 7). In detail, the Co K-edge absorption near-edge structure (XANES) spectra of the N-HP-Co SACs, as well as those of Co foil, Co₃O₄, CoO were used to compare (Fig. 2a in revised manuscript). Specifically, in detail, the near-edge absorption energy of the N-HP-Co SACs was suited between that of the Co foil and CoO; suggesting that the isolated Co atoms bear a positive charge of between the metal cobalt with zerovalent Co (0) and CoO with bivalent Co (II), indicating the N-coordinated chemical state of single Co atoms. The Fourier-transformed (FT) k³-weighted extend X-ray absorption fine

structure (EXAFS) spectrum of the N-HP-Co SACs demonstrated a main peak belonging to the Co-N scattering paths, comparing with the Co foil, no appreciable Co-Co coordination peak or other high shell peaks were detected in N-HP-Co SACs (Fig. 2b in revised manuscript). Furthermore, the coordination number of the center Co atom was about 4 according to the EXAFS fitting (Fig. 2c, d and Supplementary Fig. 7) and the mean Co-N/C bond length was 1.96 Å (Supplementary Table S2), further indicating the formed CoN₄ moiety.

Comment 2: As comparisons, the cell performance in this manuscript is among the average in the catalyst in Li-O₂ batteries. It is not fair simply compare this with commercial Pt/C catalyst. The Pt/C is not designed for Li-O₂ batteries and the authors should compare with other catalysts in Li-O₂ batteries to prove the importance and benefits of this single-atom catalyst.

Response: Many thanks for your professional suggestion. We agree with the reviewer's opinion that the cell performance in this manuscript is among the average in the catalyst in Li-O₂ batteries, which is based on the very low loading (only 0.56%) of Co single-atom on the carbon support. In light of the reviewer's suggestion, some widely used catalysts in Li-O₂ batteries such as Ru NPs, Co₃O₄ (*Nano Energy*, 2016, 28, 63–70; *ChemSusChem*, 2015, 8, 1429–1434; *J. Mater. Chem. A*, 2014, 2, 6081–6085; *ACS Catal.*, 2017, 7, 6533-6541) were synthesized and used as comparisons. The ORR and OER catalysis activity was analyzed through **the rotating disk electrode in the organic system** (*Journal of The Electrochemical Society*, 2013, 160, A344-A350) and **the full first charge–discharge curves tests**. The results and corresponding discussion are displayed in Supplementary Figs. 8, 9 and the revised manuscript (Lines 22-24, page 7; Lines 10-13, page 8; Lines 17-24, page 9). It is found that the N-HP-Co SACs exhibits a higher electrocatalytic activity compared with the other catalysts based on the same loading. In the future work, it is believed that the battery performance could be further improved according to the high-efficient preparation of the high Co content single-atom catalyst through the experimental improvement, although there are still challenges ahead.

Comment 3: The measurements in Q1 should be done on the catalysts after cycling to confirm whether the catalyst has been poisoned or not.

Response: Appreciate for the reviewer's suggestion. Considering the reviewer's suggestion, the XANES, EXAFS, X-ray absorption spectra data of the catalysts after 50th cycle have been carried out and analyzed. The results showed that N-HP-Co SACs in the cells after 50th cycle were still Co single atom, further indicating the durability of the N-HP-Co SACs. The corresponding results are shown in Fig. 6c, d and the related description in Lines 12-14, page 14 in the revised manuscript.

Comment 4: The quantification of Li_2O_2 is necessary as well after the 1st, 5th and 50th discharge and charge to measure how much Li_2O_2 has been formed and decomposed during cycling.

Response: Thanks for the reviewer's helpful guidance. According to your suggestion, the *in situ* differential electrochemical mass spectrometry (DEMS), the quantitative Gas chromatography (GC) analyze and UV-vis spectra have been carried out and analyzed, which can quantitatively confirm the Li_2O_2 amount.

The *in situ* differential electrochemical mass spectrometry (DEMS) of gaseous oxygen evolution and the Gas chromatography (GC) analyze after charging upon the 1st, 5th and 50th are carried out to prove the reversibility of the battery. And the corresponding results and discussion are displayed in Supplementary Figs. 28, 29, Supplementary Table S3 and the revised manuscript (Lines 18-25, Page 13). The DEMS results indicate that much more O_2 is formed in the cell with N-HP-Co SACs than 0.6% commercial Pt/C and less other signals (such as CO_2 , NO and H_2O) can be observed during the charging process at the end of different cycles. Based on the quantitative GC results, the charge efficiency of the N-HP-Co SACs after 1, 5, 50 cycles are 87.4%, 84.6% and 52.9%, respectively, which are much higher than that with 0.6% commercial Pt/C of 82.9%, 67.7% and 47.8% after 1, 5, 20 cycles.

The UV-vis spectra tests of discharge-charge cathodes are carried out to identify the components of the discharge-charge products and the reversibility of the battery, which is also a commonly used detection method (*Chem*, 2018, 4, 1345–1358; *Science*, 2018, 361, 777-781; *J. Phys. Chem. C*, 2017, 121, 9657-9661; *Angew. Chem. Int. Ed.*, 2017, 56, 4960–4964). The UV-vis spectra were obtained from the pristine TiOSO₄-based standard solution by adding a certain amount of H₂O₂ and the calibration curve for H₂O₂ in Ti(IV)OSO₄ was detected according to the absorbance of UV-vis curves at 405 nm. Based on these calibration curves, the UV-vis measurements of the discharged and recharged cathodes were carried out to indirectly quantify the amount of consumed/formed O₂ and the reaction products. The corresponding results and discussion are displayed in Supplementary Fig. 30 and the revised manuscript (Lines 1-5, page 14). The results indicate that the cell with the N-HP-Co SACs exhibits a higher Li₂O₂ formation efficiency (93.9%) and lower Li₂O₂ residual rate (4.8%) than the cell with the 0.6% commercial Pt/C (81.9% and 23.6%) after the first discharge and recharge, respectively; specifically, as the number of cycles increases, the difference is very obvious.

Comment 5: The PXRD quality is low and the signal and noise ratio should be optimised. For FTIR, it would be good to have the band from 4000 cm⁻¹ to 400 cm⁻¹ to demonstrate whether LiOH and other carbonates exist.

Response: Thanks for the reviewer's suggestion. As reviewer suggested that the PXRD quality for the signal-to-noise ratio is optimized and the corresponding results are displayed in Figs. 1g, 3f and Supplementary Figs. 13, 17, 20, 23. Furthermore, the FTIR spectra with a band range from 4000 cm⁻¹ to 400 cm⁻¹ (Supplementary Figs. 11, 25, 26) indicate that Li₂CO₃, HCO₂Li, CH₃CO₂Li except for LiOH are observed in the commercial Pt/C and N-HP-Co SACs catalysts loaded onto CP cathodes after continuous discharge-charge cycles because of non-obvious peak at 400 cm⁻¹ and the coincident peaks with Li₂O₂ and Li₂CO₃. More particularly, the cathodes with N-HP-Co SACs exhibit less by-products than that with the

commercial Pt/C, which is in accordance with the XPS (Fig. 5f, g, Supplementary Figs. 12 and 24) and ^1H NMR results (Supplementary Fig. 27) in the revised version.

Comment 6: How can this catalyst have effect both on discharge and charge? The authors need to clarify the mechanism with more evidence.

Response 6: Thanks for the reviewer's comment. The isolated Co single atom catalyst plays three vital roles in the Li-O₂ batteries.

(1) The superior catalytic activity of isolated Co single atom catalyst speeds up the kinetic of Li₂O₂ formation and decomposition. The charge-discharge overpotential (Fig. 3a, c and Fig. 4a in the revised manuscript and Supplementary Table S4 in the Supplementary Materials) and CV curves (Fig. 3b) of the cells demonstrate the relative superior catalysis performance of the Co SACs compared with the catalysts with the same loading. Considering the reviewer's suggestion, the current-voltage curves at 5 mV/s on a GC rotating disk electrode (900 rpm) for oxygen reduction were carried out and analyzed to further demonstrate the faster kinetic of the ORR with N-HP-Co SACs in Li-O₂ batteries. The corresponding results and discussion are displayed in the Supplementary Fig. 8 and the revised manuscript (lines 22-24, page 7).

(2) During the discharge, the homogeneous active sites of the single atom catalysts is an important factor in tuning the deposition behavior and morphology of Li₂O₂. On the cathodes with commercial Pt/C catalyst, the Li₂O₂ with toroidal morphology is also consistent with the results obtained by other groups (*ACS Catal.*, 2018, 8, 9006-9015; *Adv. Funct. Mater.*, 2016, 26, 7626-7633) (Fig. 3d in revised manuscript). In sharp contrast, the unique nanosheet Li₂O₂ are uniformly grown on the N-HP-Co SACs cathode which provide sufficient Li₂O₂-electrolyte interfaces, promote the decomposition of the products during charge and result in the rechargeability enhancement of Li-O₂ cell (Fig. 3e in revised manuscript). The growth of nanosheets that corresponds to the complex electronic structure at the CoN₄ sites where the Li⁺, electrons and O₂ occur and subsequently feed as "seeds" on the surface, then

transfer to the bottom side as Li_2O_2 . This indicates the growth process is sustained by continuous mass transport to the top of the nanosheets benefited by the directional alignment of the Co SACs (Fig. 3g, h in revised manuscript). Considering the reviewer's suggestion, to further illustrate the role of the size-controlled on the discharge products, the Li_2O_2 on cathode with the Co nanoparticles is used for comparison, and the discoid discharged products can be observed in the following Figure R1.

(3) During the recharge, the “non-sticky” surface of the single atom sites can promote Li_2O_2 decomposition through a one-electron process. Density functional theory (DFT) calculations demonstrate that the binding energies of the reduced species LiO_2 on N-HP-Co SACs surface is -1.03 eV, indicating that the “aggravic” species can easily spread to the electrolyte from the N-HP-Co SACs surface (Fig. 4c-h and Supplementary Fig. 18). This means that the decomposition of the Li_2O_2 is likely to be the one-electron routes by disproportionate reaction of LiO_2 in the electrolyte. Furthermore, the UV-vis absorbance and nitrotetrazolium blue chloride detection measurements prove that more LiO_2 is formed in the electrolyte (Fig. 4a, Supplementary Figs. 14 and 15). Considering the reviewer's suggestion, we conduct ex situ electron paramagnetic resonance (EPR) experiments of the recharged reactions to confirm the content of the intermediate LiO_2 in the electrolyte, and the corresponding results and discussion are displayed in the Supplementary Fig. 16 and the revised manuscript (lines 7-8, page 11). The EPR signal of N-HP-Co SACs after recharge can be assigned to the relatively more LiO_2 species, which is corresponding to the DFT calculations and UV-vis spectra results.

Figure R1. FESEM images of the discharged cathodes with N-HP-Co NPs at a current density of 100 mA g^{-1} with the limiting specific capacity of $3,000 \text{ mAh g}^{-1}$.

Comment 7: I do not like studies in Li-O₂ batteries where the discharge capacity claimed to be over 15000 mAh/g, while cycling at a 1000 mAh/g capacity. Why not has the full capacity cycling? I don't understand Fig. 3c, voltage vs. capacity in different rates? As the authors claimed this superior rate performance, and then why not cycling at relatively higher rates with a much bigger capacity?

Response 7: Thanks for the reviewer's comment. In fact, the cycling tests were carried out by the limited capacity to avoid a deep discharge, which is also accompanied with an associated electrode passivation. What's more, the overcharge caused by the full discharge-charge would produce more by-products leading to the poor cycling. For these reasons, the capacity limited cycling is adopted to verify the properties of the obtained material, which is also a commonly used method in this field (*Nat. Commun.*, 2019, 10, 602; *Nature*, 2018, 555, 502-506; *Science*, 2018, 361, 777-781; *Nat. Commun.*, 2017, 8, 14308; *Adv. Mater.*, 2016, 28, 857-863). In light of the reviewer's suggestion, a range of tests at relatively higher rates with a much bigger capacity are carried out. The cycle performance test of the battery at a depth of discharge of 100% is carried out, and the corresponding cycle curves and discharge/charge curves are added to the Supplementary Fig. 9d. The original Fig. 3c was replaced with the curves of

voltage vs. cycle number in different rates. The full discharge-charge curves at different current density were added to the Supplementary Fig. 9b, c and the corresponding discussion are displayed in the revised manuscript (lines 8-13, page 8). The results can also verify the advantages of N-HP-Co SACs, although there is still a less than totally satisfying with the battery cycling performances with a much bigger capacity. Because the research of Li-O₂ battery is still in its infancy, and the overall performance is very poor. The ideal electrolyte, catalyst, cathode, and corresponding theoretical recognition are in its stage of development so far, and more effort and further research are needed to develop more powerful Li-O₂ battery. In this work, the obtained promising result of proof-of-concept experiment is helpful for deepening the understanding of the effect of catalyst on discharge and charge, and providing insights in design principle of highly efficient catalysts for Li-O₂ battery, also plays a great role in broadening the future investigations.

Thank you so much again for your constructive comments to improve the quality of our manuscript.

Thank you again for your kind consideration.

REVIEWERS' COMMENTS:

Reviewer #1 (Remarks to the Author):

The authors have properly revised the manuscript and addressed my initial comments to quantify the reaction products. Therefore, I suggest this work for publication.

/Reza Younesi

Reviewer #2 (Remarks to the Author):

The authors answered all my questions and I think it can bring some new insights into the field. There are some grammar errors, other than that, I am happy with the current version.

Response to Reviewers

Sincerely send our thanks to the editor and all reviewers for the effort on this manuscript. We have responded to all the questions raised by the reviewers point by point. The responses to the reviewers' concerns are addressed as follows.

Reviewer #1

Comment:

The authors have properly revised the manuscript and addressed my initial comments to quantify the reaction products. Therefore, I suggest this work for publication.

Response: Many thanks for the reviewer's valuable suggestions in improving the quality of our manuscript and we are pleased the revised manuscript is to your satisfaction. Again we greatly appreciate your suggestion for publication.

Reviewer #2

Comment: The authors answered all my questions and I think it can bring some new insights into the field. There are some grammar errors, other than that, I am happy with the current version.

Response: We greatly appreciate the reviewer's very prompt and positive evaluation on the revised manuscript for publication. Concerning the grammar errors, we have checked the grammar point by point and also sent the manuscript to the specific English language editing affiliate of *American Journal Experts* to improve its clarity and readability. All related changes have been highlighted in **Yellow** in the revised manuscript.